# Low Levels of Vitamin D and Silent Myocardial Ischemia in Type 2 Diabetes: Clinical Correlations and Prognostic Significance

**DOI:** 10.3390/diagnostics12112572

**Published:** 2022-10-23

**Authors:** Rosario Rossi, Marisa Talarico, Alessandra Pascale, Vittorio Pascale, Roberto Minici, Giuseppe Boriani

**Affiliations:** 1Cardiology Division, Department of Biomedical, Metabolic and Neural Sciences, Policlinico di Modena Hospital, University of Modena and Reggio Emilia, Via del Pozzo, 71, 41124 Modena, MO, Italy; 2Cardiology Unit, Azienda Ospedaliera Pugliese-Ciaccio Hospital, Via Pio X 83, 88100 Catanzaro, CZ, Italy; 3Radiology Unit, Azienda Ospedaliera Pugliese-Ciaccio Hospital, Via Pio X 83, 88100 Catanzaro, CZ, Italy

**Keywords:** type 2 diabetes, serum vitamin D, silent myocardial ischemia

## Abstract

Vitamin D deficiency has a pathogenetic and prognostic role in coronary artery disease and a key role in pain transmission. Diabetic patients have a higher risk of silent myocardial ischemia (SMI) due to diabetic neuropathy. We evaluated the correlation between SMI and Vitamin D serum levels in type 2 diabetic patients and assessed whether SMI patients had a worse survival rate than their symptomatic counterpart. We enrolled 253 patients admitted in our Cardiology Unit and compared them with 50 healthy volunteers. We created three sub-groups: symptomatic MI group (125, 32.4%); SMI group (78, 25.7%), and no-MI group (50, 41.9%). 25(OH)D levels (nmol/L) were lower in the SMI group (34.9 ± 5.8) compared to those in the symptomatic MI (49.6 ± 6.1; *p* = 0.01), no MI (53.1 ± 6.2; *p* = 0.001), and control groups (62.1 ± 6.7; *p* = 0.0001). 25(OH)D levels predicted SMI in diabetic patients, with an inverted odds ratio of 1.11 (*p* = 0.01). Symptomatic MI group survival was higher than the SMI one (6-year survival rate: 83 vs. 69%; *p* = 0.01). Diabetic patients with SMI had a higher mortality risk and showed lower 25(OH)D levels than the symptomatic group. This suggests the crucial role that vitamin D has in the pathogenesis of SMI.

## 1. Introduction

Vitamin D or 25(OH)D is a fat-soluble hormone obtained either indirectly by sunlight exposure or directly through dietary sources or supplements [1]. D- hypovitaminosis is common, primarily due to a lack of sunlight exposure. Age, gender, ethnicity, skin color, season, and clothing influence vitamin D levels, too [2]. It is estimated that 1 billion people worldwide suffer from vitamin D deficiency [3]: more than 40% of the USA and European populations and even 80% of the Italian population suffer from this [4,5]. Low 25(OH)D levels were associated with cardiovascular disorders, including heart failure, stroke, and, especially, coronary heart disease (CAD) [6,7,8,9,10]. Moreover, 25(OH)D deficiency was associated with endothelial dysfunction, subclinical atherosclerosis, a significant reduction in coronary flow reserve [11,12], and sub-epicardial ischemia in patients hospitalized for the acute coronary syndrome [13]. CAD is a leading cause of death in type 2 diabetes patients. In the diabetic population, silent myocardial ischemia (SMI) is extremely common and carries a higher risk of severe complications than symptomatic MI [14,15,16,17]. SMI is the result of an ‘anginal warning system’ failure, resulting in a delay in anti-ischemic therapy [17,18]. Moreover, the pathophysiology of SMI in diabetic patients is poorly understood [19,20,21]. In line with other previous large studies that demonstrated a potential correlation between 25(OH)D deficiency and subclinical myocardial injury and sensitive neuropathy [22,23,24,25], we hypothesized that vitamin D deficiency is correlated with SMI. Among the studies reporting a strong association between low 25(OH)D levels and MI, none of them differentiated between patients with silent myocardial ischemia (SMI) and symptomatic myocardial ischemia, even if 25(OH)D may be related to one or another sub-type. There is thus a clear rationale for measuring 25(OH)D levels in carefully characterized type 2 diabetic patients with CAD and MI. The purpose of the present study is two-fold. First, we established the importance of vitamin D levels in SMI development; second, we compared the prognosis of diabetic patients with silent MI to the prognosis of diabetic patients with symptomatic MI in order to assess whether SMI patients had a worse survival rate (as expected) than their symptomatic counterparts.

## 2. Materials and Methods

### 2.1. Study Population, Setting, and Data Collected

In this single-center, cross-sectional, retrospective, and observational study, we enrolled, from 1 January 2000 and 31 December 2013, 2407 diabetic patients with suspected CAD who underwent coronary angiography at the Catheterization Laboratory of the Policlinico di Modena Hospital. We considered silent myocardial ischemia as evidence of ≥0.10 mV exercise-induced ST-segment depression at EKG (considered as highly positive) without symptoms (angina or equivalent) and as evidence of significative coronary artery disease (≥70% diameter stenosis in at least one of the main coronary branches; or ≥50% in the left main). We enrolled retrospectively patients admitted to the Cardiology Unit for suspected acute or chronic coronary artery disease who underwent a positive exercise test until 6 weeks before the admission and a vitamin D assay until 12 months from the coronary angiography. Patients with primary valvular, congenital, myocardial, and pericardial disease, as well as patients with previous surgery or percutaneous revascularization, previous acute coronary syndromes, and/or EKG pathological Q waves, were excluded. Among the eligible patients, 960 patients also had an exercise test within 6 weeks of catheterization. In this group of patients, 203 were found to have significant CAD (≥70% diameter stenosis in at least one of the main coronary branches, or ≥50% in the left main coronary branch) on coronary angiography and had ≥0.10 mV exercise-induced ST-segment depression at EKG. The last patients, together with a group of 50 patients with type II diabetes and no MI and 50 healthy, age- and gender-related non-diabetic control subjects, constituted the total selected population. All enrolled patients had at least one 25(OH)D serum determination until 12 months before catheterization in the year of catheterization. At 6 and 12 months after catheterization, and then yearly, follow-up information was obtained by clinic visits, telephone interviews, or a combination. Follow-up was obtained for 95% of the studied population.

### 2.2. Assessments of Vitamin D Status

Vitamin D is a steroidal substance that is mainly produced in the skin by direct exposure to sunlight. The principal forms of vitamin D are cholecalciferol (D3) and ergocalciferol (D2). The human body cannot produce D2, but it may be ingested in the form of supplements. Vitamin D is not as biologically active as D3. It undergoes successive hydroxylations in the liver and kidney to form the active hormone 1,25-dihydroxyvitamin D (calcitriol). 25(OH)D is vitamin D3’s major storage form, with plasma levels more than 1000-fold greater than those of the active 1,25-dihydroxyvitamin D. Hence, 25(OH)D measurement was adequate for calculating the overall vitamin D status. We used a chemiluminescence immunoassay method for the quantitative determination of total serum 25(OH)D (DiaSorin, Stillwater, MN, USA). People on vitamin D oral supplementation were excluded, as were those using sunbeds.

### 2.3. Statistical Analysis

All analyses were performed using the statistical package SPSS, version 22.0 (IBM Corp., Armonk, NY, USA). Baseline characteristics were expressed as the mean ± one standard deviation (SD) or the standard error of the mean (SEM), when specified. Categorical variables were described as percentages. One-way analysis of variance (ANOVA) was used to compare the mean of the baseline characteristics and the post hoc analysis to make comparisons between the group means; and the χ^2^ test was used for dichotomous variables. Analysis of covariance (ANCOVA) was performed to examine differences in mean serum vitamin D levels between the study groups, with age, gender, BMI, and season of vitamin D measurement, as covariates. In the logistic regression analysis for predicting the likelihood of SMI, we compared anthropometric characteristics, risk factors, and angiographic parameters. The parameters showing statistical significance were used as co-variates in the ANCOVA analysis. Direct logistic regression was executed, and odds ratios were calculated to assess the impact of various potentially independent factors on SMI. Survival probabilities were estimated using the Kaplan–Meier method. Breslow’s formulation of the Cox proportion hazard model was used to test for a significant association between survival time and the presence of SMI. Differences were considered statistically significant when the *p*-value was < 0.05.

## 3. Results

### 3.1. Demographic, Clinic, and Catheterization Characteristics of Our Study Population

Table 1 summarizes the demographic details and study assessments performed for each group. Patients with SMI had a higher body mass index (BMI) and were older compared to the patients with symptomatic MI.

The SMI group demonstrated a longer duration of diabetes, but there were no significant differences in HbA1c between them. After adjusting for age, gender, BMI, and season of vitamin D measurement, the 25(OH)D levels (nmol/L) (SE) were significantly lower in patients with SMI (34.9 ± 5.8) with respect to patients with symptomatic MI (49.6 ± 6.1; *p* = 0.01), those with no MI (53.1 ± 6.2; *p* = 0.001), and healthy volunteers (62.1 ± 6.7; *p* = 0.0001). Pairwise comparisons revealed the main group significance between patients with silent MI and those with symptomatic MI (see Table 1). Direct logistic regression was performed to assess the impact of all potential independent variables on the presence of SMI. The full model that emerged was statistically significant (χ^2^ = 27.3, *p* = 0.001).

### 3.2. Incidence of SMI

As shown in Table 2, vitamin D was the only independent variable that made a statistically significant contribution to the model, with an inverted odds ratio of 1.11 (*p* = 0.01). For each unit reduction in vitamin D, the likelihood of SMI increased by 11%. At the follow-up, most of our patients underwent coronary revascularization: 93.6% (117 of 125) of symptomatic MI patients vs. 94.9% (74 of 79) in the SMI group (*p* = 0.3, inter-group comparison).

### 3.3. Therapeutic Options and Survival

Detailed therapeutic options are shown in Table 3.

We observed that the two studied groups were comparable to the type of revascularization. The median follow-up time for the study population was 6 years. The symptomatic MI patients had a better survival rate than the SMI patients (6-year survival rate: 83 vs. 69%; *p* = 0.01; Figure 1).

It can be observed that, during the first years after revascularization, the survival curves appear approximately coupled. The divergence of the curves occurred from the third year onwards. From the univariate analysis, we recorded 24/78 deaths in the silent ischemia group. Among them, 13 patients had sudden cardiac death, 3 patients had non–cardiac death, and 8 patients had cardiac death (heart failure or acute coronary syndrome). In the angina group, we recorded 21/125 deaths. Among them, 7 patients had sudden cardiac death, 15 patients died of acute coronary artery syndrome, and 8 patients had non-cardiac death.

## 4. Discussion

Modern research revealed a new horizon and function for 25(OH)D beyond its proven role in the treatment of rickets and osteoporosis. 25(OH)D deficiency is believed to be associated with multiple sclerosis, respiratory diseases, many types of cancers, metabolic disorders, diabetes, and cardiovascular disease [7,8,9,26,27,28]. Our results demonstrated that 25(OH)D deficiency was significantly associated with SMI in our population of well-selected type 2 diabetic patients, all with significant CAD. After adjusting for age, gender, body mass index, and season of 25(OH)D measurement, we demonstrated significantly lower serum levels in patients with SMI vs. patients with symptomatic MI, those with no MI, and healthy volunteers. Pain is a distressing sensation, as well as an emotional experience that is linked to actual or potential tissue damage, with the sole purpose of notifying the mechanism to react towards a stimulus to avoid further tissue damages. The sensation of pain is associated with the activation of the receptors in the primary afferent fibers, which is inclusive of the unmyelinated C-fiber and myelinated Aσ-fiber. Because pain is mediated by small sensory fibers, our findings allow us to hypothesize that vitamin D deficiency is possibly related to the presence of SMI, the promotion of an abnormal transmission of pain, or the degenerative neuropathy of the small nociceptive nerve fibers. This view is supported by the experimental studies on the effects of 25(OH)D on the nervous system. 25(OH) D deficiency was associated with small nervous fibers neuropathy involved in the transmission of pain [22,24]. The experimental data lead to the evidence that 25(OH)D has a pivotal role in the regulation of the synthesis of neurotrophic factors, the conductance velocity of motor neurons, neuronal plasticity processes, and neuroprotective actions [29]. Vitamin D receptors are widely expressed by the central and peripheral nervous systems [30]. Particularly, in the peripheral nervous system, the vitamin D receptors are found in predominantly nociceptive neurons of the dorsal root ganglia [31]. The vitamin D receptors tend to rearrange themselves in the neurons of diabetic rats [32], allowing us to posit the role of low levels of vitamin D in the abnormal transmission of pain in diabetics. v25(OH)D has an important role in promoting nerve growth factor (NGF) secretion. NGF is a target-derived protein that regulates the phenotype and sensibility of nociceptor fibers, and its deficiency may lead to the development of clinical diabetes small fibers neuropathy [33]. A recent study has shown a positive correlation between 25(OH)D and serum NGF in diabetes patients [33,34]. A study in rodents reported the deactivation of 25(OH)D in the presence of hyperglycemia [35]. This may result in decreased vitamin D-mediated NGF secretion, which, in turn, could lead to a predominantly small nerve fibers neuropathy. In other words, it seems reasonable to assume a pathophysiological link between diabetes, low levels of vitamin D, and compromising nociceptor fibers. Several prognostic studies have shown that SMI is associated with an unfavorable outcome in clinical patients affected by acute myocardial infarction, unstable angina, and chronic stable CAD [34,35,36,37]. In type 2 diabetic patients, SMI is an issue of public health and a condition significantly related to the mortality rate [14,17,38,39]. The survival analysis in our study confirmed an increased mortality risk in SMI type 2 diabetic patients compared to patients with angina, despite the same extent of CAD and the optimal revascularization obtained. While the pathophysiology of silent myocardial ischemia is unclear, a defective anginal warning system may be the cause, which may increase the risk of subsequent cardiac events [34,35]. The absence of pain combined with ischemic episodes has been ascribed to a variety of mechanisms, including the interruption of afferent pain fibers, distal coronary stenosis, an abnormal vasomotor tone at the sites of ungrafted and minor coronary artery stenosis, and psychological factors [36]. Moreover, in patients with silent myocardial ischemia, revascularization treatment was associated with a significantly lower long-term risk of cardiac death compared with the MT-alone group, supporting the contemporary practice of ischemia-directed revascularization, even in patients with silent myocardial ischemia [37]. The silent myocardial ischemia group showed the worst survival rate in our study. The occurrence of overly mild or atypical symptoms, pushing the patient to seek medical advice or to get the attention of the healthcare providers, made the ischemia discovered later. Firstly, this condition exposes the patient to all of the myocardial infarction mechanical and electric complications (such as septal or myocardial rupture, arrythmias, acute mitral regurgitation, thrombosis, and aneurisms), due to the impossibility of receiving a prompt revascularization treatment. These acute or sub-acute complications could be the reasons for the sudden cardiac deaths reported in our results (13/24 patients died of sudden cardiac death in the SMI group, and only 7/21 died in the angina group). Moreover, several studies showed the association between heart failure and SMI, which may be another cause of death. In our study, 8/24 SMI patient deaths were attributed to heart failure and coronary artery syndromes together; in the angina group, no heart-failure related death was reported in the observational period. In our opinion, the correlation between SMI and hearth failure should be further investigated as a possible cause of mortality and morbidity in such patients.

In conclusion, we conducted a carefully designed study that involved detailed clinical and invasive/non-invasive procedures in order to accurately stratify the presence of CAD and MI in all enrolled patients. Our study included an appropriately matched disease control group (type 2 diabetic patients without MI) and healthy volunteers. The findings suggest a role of 25(OH)D in MI. Our results also suggest that low levels of 25(OH)D may contribute to the development of diabetic-related neuropathy. Further long-term prospective studies are needed to examine causality, i.e., if low 25(OH)D levels make the MI silent or if they are a risk factor/surrogate marker for the development of SMI. It may be reasonable to obtain plasma levels of 25(OH)D in all diabetic patients with significant CAD and to consider hypovitaminosis D as a factor potentially associated with SMI. It is also confirmed that SMI is a significant negative prognostic factor for mortality in type 2 diabetic patients. Whether hypovitaminosis D will represent a therapeutic target for the treatment of neuropathy or whether it will be able to improve prognosis in diabetic patients with MI will have to be evaluated in subsequent randomized trials. The current study provided a novel point of view on the role of low Vitamin D levels in SMI development in type 2 diabetes, which has never been investigated before, to the best of our knowledge. Moreover, to remove possible confounders, and considering vitamin D physiology, we excluded patients on vitamin D oral supplementation and those using sunbeds and recorded the season of vitamin D measurement. Then, we did not consider vitamin D deficiency in our analysis but rather low vitamin D levels. The reason for this is that low vitamin D levels that do not fulfill the strict definition of vitamin D deficiency may have a pathogenetic role in SMI development.

Several limitations must be recognized. Firstly, this is an observational study, so it is not possible to establish causality, only associations. An additional limitation concerns the sample size, which was not calculated but rather derived from the number of patients who matched the strict inclusion criteria and the relatively small cohort size and the cross-sectional design. To minimize the impact of the sample size, we carefully characterized and matched patients in different cohorts. Moreover, we did not study vitamin D deficiency and/or insufficiency itself, as defined in the current guidelines, but we evaluated vitamin D levels among the groups, showing a statistical correlation in survival outcome, according to our initial opinion. Another limitation is that we did not record, in the current clinical practice, details of the patients’ diet status. Moreover, since this is a retrospective study, recording such information could be very difficult. These data are not present in the current study. The only statistical significance between subgroups (patients with SMI vs. patients with angina) was the higher body mass index and age for patients in the SMI group. This was expected, as age, the duration of diabetes, and the features of metabolic syndrome (including obesity) are well-known risk factors for SMI [40]. To address this, we included age, body mass index, and duration of diabetes as covariates in the ANCOVA and still found that mean serum 25(OH)D levels were significantly lower in patients with SIM. Another limitation is that 25(OH)D plasma levels are influenced by various factors: age, gender, ethnicity, skin color, season, and clothing. To overcome this limitation, we studied whether there was an imbalance regarding the season in which the dosage of 25(OH)D was affected. The studied groups, including the healthy controls, were all statistically comparable in terms this parameter. Furthermore, summer (from July to September) 25(OH)D dosage was inserted as a covariate in the statistical analysis, together with age and gender, and these parameters did not affect the result. Moreover, all of the enrolled patients belonged to the same ethnic group and came from the same geographical area, so we postulated equal skin color and similar habits regarding sun exposure and clothing.

## 5. Conclusions

In the current study, we established the importance of vitamin D levels in SMI development in the population of type 2 diabetic patients. Based on the results of this study, the SMI group had lower levels of 25 OH vitamin D compared with other groups. Moreover, we demonstrated that SMI patients with low vitamin D levels had a worse prognosis than their symptomatic counterparts. These findings may open new horizons in the risk-stratification, management, and prevention of cardiovascular complications in type 2 diabetes. Further prospective studies that are larger are needed to better investigate and confirm the current study’s novel findings.

## Figures and Tables

**Figure 1 diagnostics-12-02572-f001:**
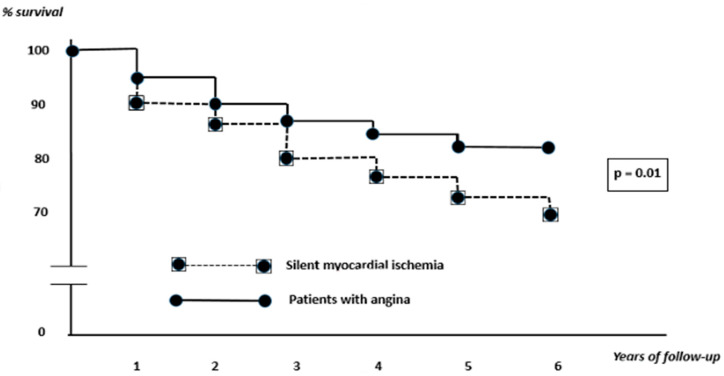
Six-year survival of diabetic patients with silent myocardial ischemia vs. symptomatic ischemia.

**Table 1 diagnostics-12-02572-t001:** Demographic, clinic, and catheterization characteristics of our study population.

Parameter	Healthy Volunteers	No Myocardial Ischemia	Symptomatic Myocardial Ischemia	Silent Myocardial Ischemia	Post Hoc *p*-Value, Comparing Silent and Symptomatic Ischemia
n	50	50	125	78	
Demographic and anthropometric parameters					
Age, years	62.0 ± 9.0	55.4 ± 8.2	61.9 ± 9.4	64.1 ± 10.3	0.03
Male gender, %	72.0 (*n* = 36)	80.0 (*n* = 40)	72.0 (*n* = 90)	71.8 (*n* = 56)	0.7
Ethnicity	Caucasian	Caucasian	Caucasian	Caucasian	
Coronary risk factors					
BMI, kg/m^2^	26.1 ± 4.6	30.1 ± 6.7	31.1 ± 5.5	33.8 ± 6.6	0.04
Diabetes duration, years		6.5 ± 6.0	13.5 ± 7.0	15.5 ± 9.0	0.03
HbA1c, mmol/mol		62 ± 16	62 ± 15	65 ± 17	0.2
HbA1c, %		7.9 ± 1.9	7.9 ± 1.6	8.1 ± 1.6	
Total cholesterol, mmol/L		4.1 ± 1.4	4.2 ± 1.5	4.0 ± 1.6	0.6
MAP, mmHg	91.1 ± 12.8	101.5 ± 11.4	100.9 ± 11.0	102.0 ± 12.7	0.1
eGFR, ml/min/1.73 m^2^	90 ± 25	82 ± 21	67 ± 23	65 ± 24	0.5
Urine ACR, mg/mmol		0.6 ± 0.9	4.1 ± 7.2	3.3 ± 30.1	0.1
25(OH)D, mmol/L (±SEM)	62.1 (6.7)	53.1 (6.2)	49.6 (6.1)	34.9 (5.8)	0.01
Summer (from July to September) determination of 25(OH)D, % (n)	28.0 (*n* = 14)	30.0 (*n* = 15)	28.8 (*n* = 36)	29.5 (*n* = 23)	0.6
Left ventricular ejection fraction, median (iqr), %	65 (60–70)	60 (50–70)	58 (50–66)	59 (52–66)	0.1
Exercise test characteristics					
Exercise time (iqr), min		8.5 (7.8–9.2)	4.5 (3.9–5.1)	4.9 (3.9–5.9)	0.1
Exercise heart rate, median (iqr), bpm		128 (112–138)	112 (105–119)	118 (109–127)	0.3
ST depression, median (iqr), mm			0.17 (0.15–0.20)	0.20 (0.12–0.28)	0.2
Catheterization characteristics					
1-vessel CAD, % (n)			20.0 (*n* = 25)	19.3 (*n* = 15)	0.9
2-vessel CAD, % (n)			29.6 (*n* = 37)	26.9 (*n* = 21)	0.4
3-vessel CAD, % (n)			47.2 (*n* = 59)	50.0 (*n* = 39)	0.5
Left main, % (n)			3.2 (*n* = 4)	3.8 (*n* = 3)	0.8

ACR indicates the albumin creatinine ratio; BMI, body mass index; CAD, coronary artery disease; eGRF, estimated glomerular filtration rate; iqr, interquartile range; MAP, mean arterial blood pressure; n, number; 25(OH)D, 25-hydroxy-vitamin D.

**Table 2 diagnostics-12-02572-t002:** Logistic regression predicting the likelihood of silent myocardial ischemia.

Parameter	*B*	SE	Wald	p	Odds Ratio	95% CI for Odds Ratio	
						Lower	Upper
Age	0.10	0.06	2.8	0.09	1.11	0.98	1.25
Body mass index	0.02	0.06	1.3	0.22	1.02	0.92	1.20
Diabetes duration	0.08	0.05	2.5	0.11	1.08	0.97	1.19
Male gender	0.27	0.40	0.9	0.8	1.25	0.51	2.87
Summer determination of 25(OH)D	0.34	0.20	1.0	0.6	1.50	0.74	2.12
25(OH)D	−0.06	0.03	5.9	0.01	1.11	1.06	1.17
Constant	−11.28	6.9	2.2	0.10	0.00		

**Table 3 diagnostics-12-02572-t003:** Therapeutic options chosen for the patients of our study, divided according to the presence of angina or silent myocardial ischemia.

Therapeutic Options *	Patients with Symptomatic Myocardial Ischemia (*n* = 125)	Patients with Silent Myocardial Ischemia (*n* = 78)	*p* **
Optimal medical treatment	3.2% (*n* = 4)	3.8% (*n* = 3)	0.5
Percutaneous revascularization	68.8% (*n* = 86)	68.0% (*n* = 53)	0.3
Surgical revascularization	28.0% (*n* = 35)	28.2% (*n* = 22)	0.8

* All patients were affected by significant coronary artery disease. ** All comparisons resulted in no statistical significance.

## Data Availability

The clinical dataset used and analyzed during the current study is available from the corresponding author on reasonable request.

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
