# Peer review of "Low Levels of Vitamin D and Silent Myocardial Ischemia in Type 2 Diabetes: Clinical Correlations and Prognostic Significance"

_diagnostics, 2022, doi:10.3390/diagnostics12112572_

Round 1

Reviewer 1 Report

This is a retrospective study of Vitamin D level correlation with silent myocardial ischemia. Generally the study looks good. However, there are several major limitations that needed to be addressed.

Major comments 

1. Please define silent myocardial ischemia. As one of your inclusion criteria was a ST depression of at least 0.1 mV. So, how patients with SMI were diagnosed and referred to to angiography or exercise tests.

2. Please explain how to select the patients to enter the study, you need to determine the power of study to support the sufficiency of your sample size.

3. Please explain how to test normality distribution of of your data. It seems there are some variables with non normal distribution of data needing to use non parametric tests to analysis.

4.As cardiovascular diseases mostly are correlated with vitamin D deficiency or insufficeny, the mean of vitamin D level in SMI is higher indicating that SMI group patients were not vitamin D deficient or insufficient.Therefore you must show the cut point level of 25 OH vitamin D that correlated with SMI.

5. Please describe diet status of patients regarding vitamin D intake.

Minor comments 

1. In the introduction section you had a wrong definition of vitamin D2. Please correct that.

2. Sum of three vessels and left main coronary patients who would need to CABG in Table1 is not equivalent to the number of patients with surgical revascularization in table 3.

Author Response

First author review

Major comments 

  1. Please define silent myocardial ischemia. As one of your inclusion criteria was a ST depression of at least 0.1 mV. So, how patients with SMI were diagnosed and referred to to angiography or exercise tests.
  2. We considered silent myocardial ischemia the evidence of ≥ 0.10 mV exercise-induced ST-segment depression at EKG (considered as highly positive) without symptoms (angina or equivalent) and with the evidence of significative coronary artery disease (≥ 70% diameter stenosis in at least one of the main coronary branches; or ≥ 50% in the left main).
  3. Please explain how to select the patients to enter the study, you need to determine the power of study to support the sufficiency of your sample size.
  4. We enrolled retrospectively patients admitted in the Cardiology Unit for suspected acute or chronic coronary artery disease who underwent to a positive exercise test until 6 weeks before the admission and a vitamin D assay until 12 months from the coronary angiography. The sample size was not calculated but derived from the number of eligible patients, because of the strict inclusion criteria.
  5. Please explain how to test normality distribution of of your data. It seems there are some variables with non normal distribution of data needing to use non parametric tests to analysis.
  6. Kolmogorov-Smirnov test and Shapiro-Wilk test were used to verify the normality assumption of data. The unpaired Student t-test was used to assess statistical differences for continuous normally distributed data, while categorical and continuous not normally distributed data were assessed using the Chi-squared test and the Mann-Whitney test, respectively.

4.As cardiovascular diseases mostly are correlated with vitamin D deficiency or insufficeny, the mean of vitamin D level in SMI is higher indicating that SMI group patients were not vitamin D deficient or insufficient. Therefore you must show the cut point level of 25 OH vitamin D that correlated with SMI.

  1. We cannot show the cut point level correlated with SMI since we should enlarge the sample size to assume that.
  2. Please describe diet status of patients regarding vitamin D intake.
  3. We didn’t record in the current clinical practice details of patients diet status. Moreover, since this is a retrospective study, recording such information could be very difficult. Page 6, line 216-217-218

Minor comments 

  1. In the introduction section you had a wrong definition of vitamin D2. Please correct that.

PAG 1 LINE 29-30 Vitamin D or 25 (OH) D is a fat-soluble hormone obtained either indirectly by sunlight exposure or directly through dietary source or supplements Instead of Vitamin D or 25 (OH) D is a fat-soluble hormone obtained either by sunlight exposure (ultraviolet B, 290-320 nm) or through dietary source and supplements, absorbed by the intestine [1].

PAGE 7 LINE 256-257  Bikle DD. Vitamin D metabolism, mechanism of action, and clinical applications. Chem Biol. 2014 Mar 20;21(3):319-29. doi: 10.1016/j.chembiol.2013.12.016. Epub 2014 Feb 13. PMID: 24529992; PMCID: PMC3968073 instead of Powers JM, Murphy JEJ. Sunlight radiation as a villain and hero: 60 years of illuminating research. Int J Radiat Biol 2019; 95: 1043-1049.  DOI: 10.1080/09553002.2019.1627440

  1. Sum of three vessels and left main coronary patients who would need to CABG in Table1 is not equivalent to the number of patients with surgical revascularization in table 3

We don’t think that the tables should be changed because I not all the patients with an indication to surgical revascularization underwent to such treatment, even after heart team discussion. Very often patients desire goes to the percutaneous option. Moreover, heart team, often take into consideration patients comorbidities, age and desire to address to an option or another. So, the fact that patients ave a left main o three vessels disease doesn’t mean that they underwent to a surgical revascularization.

Reviewer 2 Report

To the authors,

   In this single-center, cross-sectional, retrospective, observational study, Rossi R, et al. showed the significant correlation between lower serum 25(OH)D levels and the presence of silent myocardial ischemia (SMI) in patients with type 2 diabetes who underwent coronary angiography. 253 patients were included and divided into three groups (i.e., SMI; n=78, symptomatic myocardial ischemia (MI); n=125, and no MI; n=50). Serum 25(OH)D levels were lower in the order of SMI, symptomatic MI, and no MI group. Multivariate logistic regression analysis revealed significant correlation of lower 25(OH)D level and the presence of SMI. Moreover, SMI patients showed worse clinical outcome vs symptomatic MI patients in 6 years follow-up. Based on scientific background of vitamin D deficiency and small nervous fibers neuropathy in DM patient, the result of current study seems feasibility and will attract attention of cardiovascular researchers in this field. However, several remaining concerns exist which need to be addressed appropriately.

1)    (Page 2, line 63) Please explain the reason why patients with previous surgery or percutaneous revascularization, previous history of acute coronary syndrome and/or EKG pathological Q waves were excluded in this study.

2)    What was the clear definition of SMI and symptomatic MI in this study? Is the grouping determined by the exercise test? Please clarify this point.

3)    (Page 2, line 79) The sentence, “Vitamin D is not biologically active as D3.”, seems not be appropriate. Please fix it.

4)    In the logistic regression analysis to predict the likelihood of SMI, how the authors selected the candidates to include the analysis? This need to be explained.

5)    The authors showed better survival in symptomatic MI vs SMI during six-year follow up. Is the difference derived from cardiovascular cause or not? The detailed of cardiovascular death (myocardial infarction, sudden cardiac death, heart failure, etc..) should also be mentioned in the manuscript. How about the incidence of repeat revascularization? Moreover, appropriate discussion regarding why the symptomatic MI group in the current cohort showed better outcome, although they mentioned the general aspect in Page 6, line 181-182. Deeper discussion needs to be added. 

6)    Can the authors show the significant difference between initial vitamin D level and patient prognosis during 6-year follow-up? (i.e. Did fatal event (+) patients show lower 25(OH)D levels vs fatal event (-) population?) 

7)    (Page 5, line 164) “(22, 24)” should be “[22, 24]”. Please fix it.

Author Response

Second author review

To the authors,

 In this single-center, cross-sectional, retrospective, observational study, Rossi R, et al. showed the significant correlation between lower serum 25(OH)D levels and the presence of silent myocardial ischemia (SMI) in patients with type 2 diabetes who underwent coronary angiography. 253 patients were included and divided into three groups (i.e., SMI; n=78, symptomatic myocardial ischemia (MI); n=125, and no MI; n=50). Serum 25(OH)D levels were lower in the order of SMI, symptomatic MI, and no MI group. Multivariate logistic regression analysis revealed significant correlation of lower 25(OH)D level and the presence of SMI. Moreover, SMI patients showed worse clinical outcome vs symptomatic MI patients in 6 years follow-up. Based on scientific background of vitamin D deficiency and small nervous fibers neuropathy in DM patient, the result of current study seems feasibility and will attract attention of cardiovascular researchers in this field. However, several remaining concerns exist which need to be addressed appropriately.

  • (Page 2, line 63) Please explain the reason why patients with previous surgery or percutaneous revascularization, previous history of acute coronary syndrome and/or EKG pathological Q waves were excluded in this study.

Patients with previous surgery or percutaneous revascularization, previous history of acute coronary syndrome and/or EKG pathological Q waves were excluded in this study because these characteristics are at the beginning silent or symptomatic signs of ischemia.  

  • What was the clear definition of SMI and symptomatic MI in this study? Is the grouping determined by the exercise test? Please clarify this point.

We considered silent myocardial ischemia the evidence of ≥ 0.10 mV exercise-induced ST-segment depression at EKG (considered as highly positive) without symptoms (angina or equivalent) and with the evidence of significative coronary artery disease (≥ 70% diameter stenosis in at least one of the main coronary branches; or ≥ 50% in the left main). We enrolled retrospectively patients admitted in the Cardiology Unit for suspected acute or chronic coronary artery disease who underwent to a positive exercise test until 6 weeks before the admission and a vitamin D assay until 12 months from the coronary angiography. Page 2 Line 60-67

  • (Page 2, line 79) The sentence, “Vitamin D is not biologically active as D3.”, seems not be appropriate. Please fix it.

Pag 2 line 85. Vitamin D is not biologically active in the D3 form Instead of Vitamine D is not biologically active as D3.

  • In the logistic regression analysis to predict the likelihood of SMI, how the authors selected the candidates to include the analysis? This need to be explained.

In the comparison between groups (silent and symptomatic myocardial ischemia) we compared anthropometric characteristics, risk factors, and angiographic parameters. The parameters showing statistical significance were used as co-variates in the Ancova analysis. From this, the only co-variate significatively correlated to myocardial ischemia was Vitamin D status.

  • The authors showed better survival in symptomatic MI vs SMI during six-year follow up. Is the difference derived from cardiovascular cause or not? The detailed of cardiovascular death (myocardial infarction, sudden cardiac death, heart failure, etc..) should also be mentioned in the manuscript. How about the incidence of repeat revascularization?

From the univariate analysis we recorded respectively:

  • In the silent ischemia group 24/78 deaths. Over 50% of them had sudden cardiac death (13 patients); 3 patients had non – cardiac death; 8 patients had cardiac death (heart failure or acute coronary syndrome).
  • In the angina group 21/125 deaths. Only 30% of them had sudden cardiac death (7 patients); 15 patients had another coronary artery acute syndrome; 8 patients had non cardiac death.
  • Moreover, appropriate discussion regarding why the symptomatic MI group in the current cohort showed better outcome, although they mentioned the general aspect in Page 6, line 181-182. Deeper discussion needs to be added. 

While the pathophysiology of silent myocardial ischemia is unclear, a defective anginal warning sys- tem may be the cause, which may increase the risk of subsequent cardiac events (34-35). The absence of pain combined with ischemic episodes has been ascribed to a variety of mechanisms, including interruption of afferent pain fibers, distal coronary stenosis, abnormal vasomotor tone at the sites of ungrafted, minor coronary artery stenosis and psychological factors (36). Moreover, in patients with silent myocardial ischemia, revascularization treatment was associated with significantly lower long-term risk of cardiac death compared with the MT alone grou, supporting contemporary practice of ischemia-directed revascularization, even in patients with silent myocardial ischemia (37).  

  • Can the authors show the significant difference between initial vitamin D level and patient prognosis during 6-year follow-up? (i.e. Did fatal event (+) patients show lower 25(OH)D levels vs fatal event (-) population?) 

The mechanism is that lower vitamin d doses are related to a higher silent myocardial ischemia, and patients affected by such condition are more likely to die, which is the thesis of the current study.

 7)    (Page 5, line 164) “(22, 24)” should be “[22, 24]”. Please fix it.

Ok, see the new version of the manuscript

Reviewer 3 Report

It is an interesting manuscript. Authors succeed to present their data in a clear way adding information to the existing literature. Therefore, I have no corrections to do and the manuscript can be published unaltered.

Author Response

Thank you. 

Round 2

Reviewer 1 Report

1-The author answered some comments. However, the critical and fundamental issue still remains. The study title is "Vitamin D low-levels and silent myocardial ischemia in type 2 diabetes: clinical correlations and prognostic significance". Also, in the introduction section, the authors mentioned that "we hypothesized that vitamin D deficiency is correlated to SMI." The mean ± SD for 25 (OH) vitamin D is 34.9 ± 5.8 in the SMI group indicating a normal status of vitamin D.  Low level of vitamin D usually is defined vitamin D levels under 20 ng/ml. The study hypothesis, topic, and conclusion are contrary to the results. Authors need to modify the manuscript based on the above-mentioned issue and must answer comments 4 and also 2. 

2- Page 2, line 88: "The major storage form of vitamin D is D2 (25- 88 hydroxyvitamin D [25(OH)D])." . 25(OH)D is vitamin D3's major storage form. Please correct it

Author Response

Author's Reply to the Review Report (Reviewer 1)

Dear Reviewer 1, thanks for your observations. We’ll try to answer and explain.

1-The author answered some comments. However, the critical and fundamental issue still remains. The study title is "Vitamin D low-levels and silent myocardial ischemia in type 2 diabetes: clinical correlations and prognostic significance". Also, in the introduction section, the authors mentioned that "we hypothesized that vitamin D deficiency is correlated to SMI." The mean ± SD for 25 (OH) vitamin D is 34.9 ± 5.8 in the SMI group indicating a normal status of vitamin D.  Low level of vitamin D usually is defined vitamin D levels under 20 ng/ml. The study hypothesis, topic, and conclusion are contrary to the results. Authors need to modify the manuscript based on the above-mentioned issue and must answer comments 4 and also 2. 

Data from observational studies suggest that low levels of 25(OH)D can negatively affect CV health

(Lee J.H., O'Keefe J.H., Bell D., Hensrud D.D., Holick M.F. Vitamin D deficiency. An important, common, and easily treatable cardiovascular risk factor? J Am Coll Cardiol. 2008;52:1949–1956. doi: 10.1016/j.jacc.2008.08.050.). Serum 25(OH)D levels >30 ng/ml are considered vitamin D optimal levels for muscolo-scheletal health (Holick M.F., Binkley N.C., Bischoff-Ferrari H.A., Gordon C.M., Hanley D.A., Heaney R.P. Evaluation, treatment, and prevention of vitamin D deficiency: an endocrine society clinical practice guideline. J Clin Endocrinol Metab. 2011;96:1911–1930. doi: 10.1210/jc.2011-0385.; Pludowski P., Holick M.F., Grant W.B., Konstantynowicz J., Mascarenhas M.R., Haq A. Vitamin D supplementation guidelines. J Steroid Biochem Mol Biol. 2018;175:125–135. doi: 10.1016/j.jsbmb.2017.01.021.; Hintzpeter B., Mensink G.B.M., Thierfelder W., Müller M.J., Scheidt-Nave C. Vitamin D status and health correlates among German adults. Eur J Clin Nutr. 2008;62:1079–1089. doi: 10.1038/sj.ejcn.1602825.; Bischoff-Ferrari H.A., Giovannucci E., Willett W.C., Dietrich T., Dawson-Hughes B. Estimation of optimal serum concentrations of 25-hydroxyvitamin D for multiple health outcomes. Am J Clin Nutr. 2006;84:18–28. doi: 10.1093/ajcn/84.1.18.).

Currently, however, there is no consensus regarding the optimal level of the vitamin for possible preventive CVD benefits.

The point is that in our study population, not vitamin D deficiency itself, but that vitamin D low levels are related firstly to SMI and hence to a worst survival. After adjusting for age, gender, BMI, and season of vitamin D measurement, 25(OH)D levels (nmol/l) (SE) were significantly lower in patients with SMI (34.9 ± 5.8) respect to patients with symptomatic MI (49.6 ± 6.1; p = 0.01); no MI (53.1 ± 6.2; p = 0.001). In other words, we observed vitamin D levels in the two groups and demonstrated that in our study population the silent myocardial ischemia group, the one with the worst survival, showed lower vitamin D levels with respect to the symptomatic group. These Vitamin D ‘low’ levels do not necessarily correspond to a real deficiency (defined by the current guidelines). The study hypothesis is that low levels of vitamin d and not vitamin D deficiency, are related to SMI and hence to a worst survival. If You suggest to better clarify this point, we could review some passages of the text, in the meantime, we added a clarification in the ‘limitation of the study section page 7, lines 253-256.

We replied to previous comments ‘4’ and ‘2’ in the previous document. 

2- Page 2, line 88: "The major storage form of vitamin D is D2 (25- 88 hydroxyvitamin D [25(OH)D])." . 25(OH)D is vitamin D3's major storage form. Please correct it

Thank you. Corrected in line 87, page 2

Reviewer 2 Report

To the authors,

The authors answered to my questions, however, the way of answering seems not polite and not acceptable for me. Please see the followings.

1)    The answer from the authors, “Vitamin D is not biologically active in the D3 form Instead of Vitamine D is not biologically active as D3.”, does not make any sense as a sentence. Please change the explanation appropriately.

2)    In the rebuttal letter, the authors explained the detail of deaths in the current cohort during 6-year follow-up. The information also needs to be added in the main manuscript.

3)    The method for selecting the candidates that were included in the logistic regression analysis also need to be described in the main document.

4)    Regarding my question, “Moreover, appropriate discussion regarding why the symptomatic MI group in the current cohort showed better outcome, although they mentioned the general aspect in Page 6, line 181-182. Deeper discussion needs to be added. “, where the authors added the further discussion? It is not acceptable for me.

5)    My question raised last time, “Can the authors show the significant difference between initial vitamin D level and patient prognosis during 6-year follow-up? (i.e. Did fatal event (+) patients show lower 25(OH)D levels vs fatal event (-) population?)”, have not been answered appropriately by the authors. I need more explanation. 

Author Response

Author's Reply to the Review Report (Reviewer 2)

Dear Reviewer 2, thanks for your observations. We’ll try to answer and explain our thoughts and improve our work at the best. 

  • The answer from the authors, “Vitamin D is not biologically active in the D3 form Instead of Vitamine D is not biologically active as D3.”, does not make any sense as a sentence. Please change the explanation appropriately.

We intend to explain that, in the main text, we inserted the sentence ‘Vitamin D is not biologically active in the D3’ and deleted ‘Vitamine D is not biologically active as D3’.

  • In the rebuttal letter, the authors explained the detail of deaths in the current cohort during 6-year follow-up. The information also needs to be added in the main manuscript.

Thanks for the suggestion, we added in page 5 line 150-155 the details of deaths. This is an important information and adds value to the work results.

  • The method for selecting the candidates that were included in the logistic regression analysis also need to be described in the main document.

Thanks for the suggestion, we added in Page 3 Line 102-105.

  • Regarding my question, “Moreover, appropriate discussion regarding why the symptomatic MI group in the current cohort showed better outcome, although they mentioned the general aspect in Page 6, line 181-182. Deeper discussion needs to be added. “, where the authors added the further discussion? It is not acceptable for me.

Thanks for the suggestion. Regarding your last question we added in the main text - Page 6 Line 202-211 – ‘While the pathophysiology of silent myocardial ischemia is unclear, a defective anginal warning system may be the cause, which may increase the risk of subsequent cardiac events (34-35). The absence of pain combined with ischemic episodes has been ascribed to a variety of mechanisms, including interruption of afferent pain fibers, distal coronary stenosis, abnormal vasomotor tone at the sites of ungrafted, minor coronary artery stenosis and psychological factors (36). Moreover, in patients with silent myocardial ischemia, revascularization treatment was associated with significantly lower long-term risk of cardiac death compared with the MT alone group, supporting contemporary practice of ischemia-directed revascularization, even in patients with silent myocardial ischemia (37).’

We are going to add in page 6 line 211-225:

‘Silent myocardial ischemia group showed the worst survival in our study. The occurrence of too mild or atypical symptoms to push the patient to seek medical advice or to get the attention of the healthcare providers made the ischemia later discovered. Firstly, this condition exposes the patient to all the myocardial infarction mechanical and electric complications (such as septal or myocardial rupture, arrhythmias, acute mitral regurgitation, thrombosis, aneurysms), due to the impossibility to receive a prompt revascularization treatment. These acute or sub-acute complications could be the reasons for the sudden cardiac deaths reported in our results (13/24 patients died of sudden cardiac death in the SMI group and only 7/21 in the angina group). Moreover, several studies showed the association between heart failure and SMI, which may be another cause of death. In our study, 8/24 of SMI patients' deaths were attributed to heart failure and coronary artery syndromes together, instead in the angina group, no heart-failure-related death was reported in the observational period. In our opinion, SMI and heart failure correlation should be further investigated as a possible cause of mortality and morbidity in such patients.’ 

  • My question raised last time, “Can the authors show the significant difference between initial vitamin D level and patient prognosis during 6-year follow-up? (i.e. Did fatal event (+) patients show lower 25(OH)D levels vs fatal event (-) population?)”, have not been answered appropriately by the authors. I need more explanation. 

We are sorry if the explanation did not result appropriate or complete. What we wanted to explain is that we didn’t insert the analysis of the mentioned association (vitamin D levels and fatal events) in our study, even if we analyzed this aspect, for several reasons. Firstly, if we analyze the vitamin D serum levels difference between the death and the survivors' groups, we found a lower vitamin D level in the group of the death (37.6 +/-6.7 vs 45.5 +/- 6.8). However, in the Cox multivariate analysis, vitamin D levels did not result significantly as an independent parameter affecting mortality. On the contrary, from the same multivariate analysis, age, SMI, and diabetes (as expected) are independent and significant variables affecting mortality at 6 years of follow-up. Then, we think that vitamin D low levels are involved in the pathogenesis of silent myocardial ischemia and that SMI itself is the independent predictive variable of death in our population. Vitamin D is lower in SMI than in the angina group and patients affected by SMI showed the worst prognosis, not because they have lower vitamin D levels but because they have SMI whose pathogenesis is related to vitamin D levels, according to our hypothesis and at the best of our knowledge from the literature.

Round 3

Reviewer 1 Report

Please add this to conclusion " based on the results of this study, the SMI group had a lower level of 25 OH vitamin D compared with other groups"

Author Response

Dear reviewer, 

we added the suggested sentence in page 7 line 277 - 278. We are grateful for Your observations which made our work more complete and clear. 

Reviewer 2 Report

To the authors,

I feel the manuscript has been improved. The current form is acceptable for me. I have no further concerns.

Author Response

Dear reviewer, 

We are grateful for Your observations which made our work more complete and clear.